# The Assessment of Skin Homeostasis Changes after Using Different Types of Excipients in Healthy Individuals

**DOI:** 10.3390/ijerph192416678

**Published:** 2022-12-12

**Authors:** Abraham Ordoñez-Toro, Trinidad Montero-Vilchez, José Muñoz-Baeza, Raquel Sanabria-De-la-Torre, Agustin Buendia-Eisman, Salvador Arias-Santiago

**Affiliations:** 1Dermatology Department, Faculty of Pharmacy, University of Granada, 18001 Granada, Spain; 2Dermatology Department, Virgen de las Nieves University Hospital, 18012 Granada, Spain; 3Instituto de Investigación Biosanitaria ibs.GRANADA, 18002 Granada, Spain; 4Dermatology Department, Faculty of Medicine, University of Granada, 18001 Granada, Spain

**Keywords:** excipient, topical drugs, Vaseline, petrolatum jelly, W/O emulsion, O/W emulsion, foam, Beeler base

## Abstract

Excipients are used as vehicles for topical treatments; however, there are not many studies that evaluate the impact of different excipients themselves. The aim of this research is to assess skin homeostasis changes in healthy individuals after using water/oil (W/O), oil/water (O/W), Beeler base, foam and Vaseline excipients. A within-person randomized trial was conducted that included healthy individuals without previous skin diseases. Skin barrier function parameters, including stratum corneum hydration (SCH), transepidermal water loss (TEWL), pH, temperature, erythema, melanin and elasticity (R0, R2, R5 and R7), were measured on the volar forearm before and after using each excipient. Sixty participants were included in the study, with a mean age of 32 years. After applying w/o excipient erythema decreased by 25 AU, (*p* < 0.001) and elasticity increased by 6%. After using the o/w excipient, erythema decreased by 39.36 AU (*p* < 0.001) and SCH increased by 6.85 AU (*p* = 0.009). When applying the Beeler excipient, erythema decreased by 41.23 AU (*p* < 0.001) and SCH increased by 15.92 AU (*p* < 0.001). Foam and Vaseline decreased TEWL and erythema. Excipients have a different impact on skin barrier function. Knowing the effect of excipients on the skin could help to develop new topical treatments and help specialists to choose the best excipient according to the pathology.

## 1. Introduction

The skin is the largest organ in the human body. It protects against external influences such as chemicals or pathogenic microorganisms and is composed of the following three layers: epidermis, dermis and hypodermis [1]. The epidermis is the most superficial and is responsible in part for skin color, hydration and texture. The stratum corneum (SC) is the outermost part of the epidermis and strongly opposes the passage of drugs through the skin. It is composed of cells without nuclei called corneocytes, cells that only retain keratin filaments embedded in a filaggrin matrix. These cells come from keratinocytes situated in the basal layer of the epidermis and gradually lose their cytoplasmatic organelles and nucleus as a consequence of their maturation [2,3]. SC has a variable thickness depending on the anatomical region (between 10 and 50 μm) and is composed of proteins (40%), lipids (20%) and water (40%) [4,5].

The SC has a mixture of small water-soluble compounds called natural moisturizing factor (NMF) that provide hygroscopic properties, forming a barrier to prevent transepidermal water loss (TEWL) [6]. TEWL is one of the most important parameters to measure skin barrier function. It is defined as the volume of water that diffuses from the epidermis and dermis through the SC and skin surface [7]. Higher TEWL values are associated with skin barrier impairment [8]. Moreover, SC hydration (SCH) is also an important parameter to assess skin barrier function since it shows water content in the skin. Lower SCH values are associated with dermatological conditions [7]. Some research has recently observed that lower SCH values are directly associated with atopic dermatitis and disease severity [9]. Nevertheless, SCH could also increase when the stratum corneum is thin. Elasticity is another interesting parameter related to skin aging and homeostasis. Some parameters that evaluate elasticity are R0—which shows skin distensibility with implications for its firmness; R2—the ratio of the ability of skin recovery to its final distension, the closer to 100% the more elastic the skin is; R5—a ratio that represents elastic recovery after its deformation, the closer to 100% the more elastic the skin is; R7—a ratio that is referred to as biological elasticity; the closer to 100% the more elastic the skin is [10]. The epidermis is partly responsible for skin color, but the dermal fiber network and the perfusion are also very relevant in Fitzpatrick skin types 1 to 3. The reddening of the skin as a result of an external stimulation, immunologic response with or without hypersensitivity to an allergen, or a viral infection is known as skin erythema [11]. Another important parameter in skin physiology is melanin, which is found in melanosomes. It has photo-screening effects as it absorbs near-infrared, visible light and UV radiation, with absorption increasing as wavelengths get shorter. For this reason, melanin is considered a protective molecule from harmful radiation. Skin pH is also a key factor for skin barrier function, SC integrity and antimicrobial defense. Some research suggests that the deacidification of SC comprised barrier homeostasis and its integrity [12]. Other parameters, such as temperature and pH, are used to determine skin homeostasis too. Figure 1 summarizes skin barrier function parameters that can be assessed non-invasively on the skin.

Topical treatments are applied directly to the skin. Its absorption depends on the following three elements: the active ingredient, the vehicle and the skin itself. A correct skin homeostasis and hydration play an important role in maintaining aesthetic and physiological conditions that facilitate drug penetration through the skin. However, the vehicle is the most important agent that impacts the active ingredient penetration [4].

Topical formulations are usually differentiated depending on the purpose of their design. Some are made for systematic absorption (transdermal formulations), others diffuse into the skin compartments (topical formulations) and others remain on the skin surface (cosmetic products, sun creams and so on) [13]. At equal active ingredient, dose and concentration, potency increases in the following order: foams, emulsions O/W, emulsions W/O and Vaseline. A foam is an (o/w or w/o) emulsion containing mainly lipophilic substances such as hydrocarbons or waxes. They can produce dryness. For emulsions O/W, in which Beeler bases are included, contain more water than emulsions W/O. After their application, the water contained evaporates quickly, producing a brief refreshing action that is suitable for cases with acute inflammation and exudation or intertriginous lesions. On the other hand, emulsions W/O contain a large amount of oil and a very little amount of water and are therefore difficult to remove with water. They are suitable when the skin needs lubrication or hydration, such as chronic, dry lesions. Finally, Vaseline (petrolatum) has almost exclusively oily components due to their occlusive and moisturizing action, making the skin more flexible and contributing to a better penetration of most active ingredients. They are indicated for dry or very dry, scaly skin and are ideal for softening scales and crusts [4].

In short, the topical formulation chosen is as important as the active ingredient itself since it determines the effectiveness of the treatment. Knowing how the vehicles themselves act on skin homeostasis would be essential for treating patients properly. So, the objective of this study is to evaluate the impact of different vehicles on skin barrier function in healthy individuals.

## 2. Materials and Methods

### 2.1. Design

A within-person randomized trial was accomplished to assess skin homeostasis changes after five vehicles on healthy individuals’ skin (Appendix A). A within-person design was used to avoid inter-induvial variability. There are several factors that could impact on skin barrier, including sex, age or skincare habits and using a within-person design the effect of the vehicle is tested in the same person, avoiding confusion bias depending on the individuals.

### 2.2. Study Sample

Healthy individuals were recruited between January and March 2022 at the Dermatology Service of the Hospital Universitario Virgen de las Nieves in Granada, Spain and from people willing to participate in the study.

#### 2.2.1. Inclusion Criteria

Healthy individuals aged between 18 and 65 years old.

#### 2.2.2. Exclusion Criteria

Patients having a diagnosed skin disease at the moment of the study, including inflammatory skin diseases such as psoriasis, hidradenitis or atopic dermatitis or any other kind of disease that may alter the epidermal barrier function and skin homeostasis;Patients who are undergoing topical, physical or systemic treatment for any disease that may alter epidermal barrier function and skin homeostasis;Failing to sign the informed consent form.

### 2.3. Excipients Used and Measurement Location

The following five vehicles were used: Vaseline, foam, Beeler base and emulsion W/O and O/W. Vaseline was composed of “VASELINA FILANTE” (211997) 50 g from ACOFARMA^®^. Foam was composed of “ESPUMAX” (2021-175) 50 mL from JOSE MESTRE, S.A. Beeler base was composed of “BASE BEELER” (200855-S-1) 50 g from ACOFARMA^®^. Emulsion W/O was composed of “COLD CREAM” (200481-Q-1) 50 g from ACOFARMA^®^. Emulsion O/W was composed of “BASE EMULSION NEO PCL O/W (SE) (008732) from Laboratorios Santamaría.

The volar forearm of each individual was divided into the following 6 areas: one for Vaseline, another for foam, Beeler, emulsion w/o and o/w. The distance between each was 2 cm, enough distance to ensure that there were no interactions between bases. One area was left untreated (control area), Figure 2 and Figure 3.

### 2.4. Variables

Homeostasis parameters associated with epidermal barrier function were measured. SCH (measured in arbitrary units, using Corneometer^®^ CM 825, Courage + Khazaka electronic GmbH, Köln, Germany), skin temperature (measured in °C, using Skin- Thermometer ST 500, Courage + Khazaka electronic GmbH, Köln, Germany), pH (measured in pH units, using Skin-pH-Meter PH 905, Courage + Khazaka electronic GmbH, Köln, Germany), TEWL (measured in g·m^−2^·h^−1^, using Tewameter^®^ TM 300, Courage + Khazaka electronic GmbH, Köln, Germany), erythema index (in arbitrary units, using Mexameter^®^ MX 18, Courage + Khazaka electronic GmbH, Köln, Germany), melanin index (measured in arbitrary units, using Mexameter^®^ MX 18, Courage + Khazaka electronic GmbH, Köln, Germany) and skin elasticity (using R0 value measured in millimeters, R2, R5 and R7 value measured in %, using Cutometer^®^ Dual MPA 580, Courage + Khazaka electronic GmbH, Köln, Germany) were measured by Multi Probe Adapter (MPA, Courage + Khazaka electronic GmbH, Köln, Germany). Elasticity parameters were measured two times while the rest of parameters were measured ten times, using their mean for analysis. Skin homeostasis parameters were measured in the volar area of the forearm on all areas previously described. The measurements were performed in the same room at Hospital Virgen de las Nieves, Granada, Spain. The average temperature was 19 ± 2 °C and average humidity was 42 ± 2%, respectively.

Gender, sex, age, occupation, normal residence, studies, alcohol intake, smoking habit, skin hydration habit and photoprotector application habit data were collected through a clinical interview. Finally, phototype was measured by a dermatologist using Fitzpatrick phototyping scale.

### 2.5. Sample Size

Accepting an alpha risk of 0.05 and a beta risk of 0.2 in a two-sided test, 60 subjects are necessary to recognize as statistically significant a difference greater than or equal to 0.05 units. The standard deviation is assumed to be 5.5.

### 2.6. Statical Analysis

Categorical data were conducted using absolute values whereas continuous values were expressed with mean and standard deviation (SD). Student’s test or Mann–Whitney tests were performed to observe whether there were significant differences before excipients appliance and after (Appendix A). Friedman test and ANOVA followed by an α-adjusted post hoc Bonferroni test were also performed to show whether there were differences between excipients themselves. A *p* < 0.05 was considered statically significant. Statical analyses were conducted with SPSS package (SPSS for Apple MacOS, Version 25.0 Chicago: SPSS Inc., Chicago, IL, USA).

## 3. Results

### 3.1. Sample Characteristics

A total of 60 healthy individuals were included in this study, being 55% women (33/60) and 45% men (27/60). The mean age was 31.82 years old (±14.25) and 16.7% have a smoking habit. The sociodemographic characteristics of the sample are shown in Table 1. We did not find any differences in skin barrier function regarding sociodemographic characteristics.

### 3.2. Skin Homeostasis Changed after Bases Were Applied Compared to Control

Changes in skin barrier function after applying vehicles are shown in Figure 4, Appendix A. After applying Vaseline, some parameters changed significantly. TEWL significantly decreased by 3.85 g·m^−2^·h^−1^ (11.28 vs. 7.38 g·m^−2^·h^−1^, *p* = 0.002), erythema decreased by 41 AU (223 vs. 182.63 AU, *p* < 0.001) and SCH also decreased by 27.60 AU (45.40 vs. 17.81 AU, *p* < 0.001). Whereas melanin increased by 26.45 AU (103.45 vs. 190.9 AU, *p* = 0.01), R2 increased by 9.7% (65.94 vs. 75.65 %, *p* < 0.001). Regarding elasticity parameters, R5 increased by 11.4% (54.64 vs. 65.99%, *p* < 0.001) and R7 increased by 10.8% (44.86 vs. 55.70%, *p* < 0.001). Other parameters such as temperature, pH and R0 did not show any significant difference.

After applying the foam, some parameters changed significantly, as represented in Figure 2 and Appendix A. TEWL significantly decreased by 3.83 g·m^−2^·h^−1^ (11.27 vs. 7.41 g·m^−2^·h^−1^, *p* = 0.011), erythema decreased by 21.99 AU (223.62 vs. 201.93 AU, *p* < 0.001). Whereas melanin increased by 22.98 AU (103.45 vs. 126.43 AU, *p* = 0.041), R2 increased by 5.4% (65.64 vs. 71.33%, *p* < 0.001), R5 increased by 10.7% (54.64 vs. 65.32%, *p* < 0.001) and R7 increased by 10.4% (44.86 vs. 55.28%, *p* < 0.001). Other parameters such as temperature, pH, SCH and R0 did not show any significant difference.

After applying the Beeler base, some parameters changed significantly represented in Figure 2 and Appendix A. Erythema significantly decreased by 41.22 AU (223.62 vs. 182.39 AU, *p* < 0.001). Whereas SCH increased by 15.91 AU (45.40 vs. 65.31 AU, *p* < 0.001), R5 increased by 11.2% (54.64 vs. 65.82%, *p* = 0.011) and R7 increased by 7.2% (44.86 vs. 52.03%, *p* < 0.001). Other parameters such as TEWL, temperature, pH, melanin and R0 and R2 did not have a significant difference.

After applying the O/W base, some parameters changed significantly, as represented in Figure 2 and Appendix A. Erythema significantly decreased by 35.35 AU (223.62 vs. 184.27 AU, *p* < 0.001).

Whereas SCH increased by 6.85 AU (45.40 vs. 52.25 AU, *p* = 0.009), R5 increased by 6.5% (54.64 vs. 61.15%, *p* = 0.002) and R7 increased by 4.8% (44.86 vs. 49.66%, *p* = 0.004). Other parameters such as TEWL, temperature, pH, melanin and R0 and R2.

After applying the W/O base, some parameters changed significantly, as represented in Figure 2 and Appendix A. Erythema significantly decreased by 25 AU (223.62 AU vs. 198.62 AU, *p* < 0.004). Whereas R5 increased by 6.3% (54.64 vs. 60.95%, *p* = 0.003), R7 increased by 6% (44.86 vs. 50.87%, *p* < 0.001). Other parameters such as TEWL, temperature, pH, melanin, SCH and R0 and R2 did not show any significant difference.

### 3.3. Differences in Skin Barrier after Applying Different Bases

The changes in TEWL were different when applying each emollient. Vaseline and foam showed a greater decrease than W/O and Beeler in TEWL. No differences were found between foam and Vaseline. Regarding melanin, Vaseline and foam showed a greater increase than W/O, but no significant differences were found between O/W, Beeler, foam and Vaseline. Beeler, O/W and Vaseline showed a greater decrease in erythema than W/O and foam, while no differences were found between Beeler, O/W and Vaseline. Concerning SCH, Beeler and O/W showed a greater increase than W/O, foam and Vaseline; however, Beeler had the greatest increase, even showing differences with O/W excipient (*p* < 0.001). Whilst Vaseline showed the greatest decrease than W/O, O/W, Beeler and foam (*p* < 0.001). Regarding elasticity, Vaseline showed a greater increase in R2 than W/O, O/W, Beeler and foam. The foam also showed a greater increase than O/W. Vaseline and foam showed a greater increase in R7 values than W/O (*p* = 0.019, *p* = 0.02; respectively) and O/W (*p* < 0.001, *p* < 0.001; respectively), but there were no differences between Vaseline, foam and Beeler.

## 4. Discussion

There are some relevant results that must be discussed in the data presented before. Vaseline and foam decreased TEWL, whilst the Beeler excipient substantially increased SCH. All excipients decreased erythema and increased R5 and R7.

Vaseline is an excipient whose composition is almost exclusively oils that confer occlusive and emollient properties. In cases of occlusion, Vaseline decreased TEWL, which is expected since creating an oily layer prevents diffusional water loss [7,14]. Moreover, this is in agreement with previous reports that reported Vaseline decreased TEWL on healthy skin and lesioned skin and in tattoo areas. According to previous research, oily vehicles such as Vaseline can penetrate SC and help in its restoration. Vaseline helps remoisturizing the skin by starting skin barrier repair, the onset of dermal-epidermal skin moisture diffusion, and intracellular lipid synthesis [15]. The decreases in SCH could be due to Vaseline itself affecting the Corneometer since, when measures were taken, there was some residual Vaseline that was not evaporated. Dobrev explained that unabsorbed creams could interfere with the performance of experiments in which electrical skin properties are measured [16]. Other studies suggest that Vaseline application does not affect erythema in animal models such as rats, mice or minipigs [17]. However, it increases reddening in rabbits, which are widely used in preclinical trials of pharmaceutical and cosmetic creams [17,18]. We observed that erythema decreased after Vaseline application for its anti-inflammatory properties. In the case of melanin, although there were not many articles on the subject, some articles used petrolatum as a control for other cream studies. In some, control Vaseline showed a melanin increase, as in this study [19]. This could be explained by the color of the product. Nevertheless, melanin is a secondary parameter that is not particularly relevant.

Foam is the easiest vehicle to spread, and it has the lowest density. Foam decreased TEWL significantly, but it did not increase SCH values. Decreases in TEWL values could have been caused by its fast absorption through SC, which may have prevented diffusional water loss. Moreover, in patients with atopic dermatitis, foam used on dry skin also decreased TEWL [20]. SCH did not increase after foam application, which could be due to its low density, which did not confer enough hydration to be significant. Besides this, other research suggests that foam does not have enough hydration power [21]. For erythema, foam decreased it as all excipients did, which could be due to its rapid evaporation of this volatile vehicle that also may contribute to reducing stinging [22].

O/W excipients are the most popular for moisturizers used [15] and are best suited for hydrophilic drugs [21]. It is composed of an oil-in-water emulsion that can be used to produce consistent creams or lotions, which evaporate rapidly, providing a cooling effect [23,24]. O/W showed a significant increase in SCH as it hydrated the skin. According to previous research, O/W did not show a large hydration [24], however, it provided hydration although lower than ointment excipients [21]. For erythema, O/W decreased values that could be due to its hydration for SC. Finally, for R5, R7 has an increase, which means a better elasticity of the skin. In other research, O/W emulsions did not show the same path as they did not show enough significance [16].

Beeler base is considered an anionic O/W emulsion, which can be used as an excipient for lightening agents and as an alternative to Lanette excipient; however, it cannot be used for cationic nor acidic principal actives [25]. Beeler excipient showed the greatest increase for SCH. As an O/W emulsion, it provides hydration to the SC but is lower than ointments, as mentioned above. Other studies suggest that active ingredients such as hypericin using a Beeler base are highly effective in getting the active ingredient absorbed [26]. For erythema, the Beeler excipient decreased erythema, which could be due to its potential hydration. Moreover, the use of compounded melatonin in Beeler cream was effective for preventing and treating post-radiation dermatitis [27]. Finally, for R5, R7 values increased, which could be due to its higher hydration values as in recent research has studied how hydration relates to elasticity [16].

Finally, W/O emulsion is the excipient that showed the least changes compared to the control. It is widely used for lipophilic drugs [20]. The erythema decreased probably because it is an excipient that, although it did not increase the SCH, probably hydrated the skin enough to reduce the redness. Finally, for R5 and R7, this study showed an increase in their values; however, in other research, W/O emulsions did not increase elasticity [16].

Using these data collected in the study, it can be discussed how these excipients could be used in the development of future topical treatments and how actual creams are affected. In this context, we will have to consider the long-term effects of these products. Vaseline could be used as an occlusive cream as it creates an oily layer that prevents water loss for example in pruritus or dryness. Moreover, Vaseline was found to increase stratum corneum thickness and to reduce T cell infiltration in the skin for atopic dermatitis [27,28] and also was recently found that Vaseline use nasally could prevent patients from COVID-19 infections [29]. Foam also decreases TEWL, which could be due to rapid absorption [30] but not to its occlusive property since foam rapidly evaporates. Moreover, foam is more desirable for patients as it is easier to extend through sensitive or inflamed areas than excipients such as Vaseline or other emulsions [20]. In the case of the Beeler excipient, oily drops in the aqueous phase are rapidly absorbed by the skin, which increases skin hydration, whereas the aqueous phase evaporates, generating a cooling effect. Beeler excipient could help synergistically active ingredients to hydrate the skin. The same occurs for O/W, which also increases SCH but less than the Beeler base. All excipients decreased erythema; therefore, all of them could be used in creams to reduce inflammation, for example for sunburns or dermatological diseases such as acne or psoriasis. Beeler, O/W and Vaseline excipients showed a greater decrease than the others, which could be due to their hydration properties on the skin, which would reduce reddening. For pH, any excipients showed any significance; hence, its acid properties were not changed. The skin acid mantle is a key factor for SC integrity and antimicrobial defense. Some research suggests that deacidification of SC comprised barrier homeostasis and its integrity and excipients do not alter this beneficial property of the skin [12]. Finally, for elasticity parameters, some studies argued that R2 and R7 were the best parameters to assess skin elasticity and aging. R2 (Ua/Uf) values are the gross elasticity of the skin, including viscous deformation and R7 (Ur/Uf) values are biological elasticity [31]. Some argued that R7 had the strongest skin aging correlation than others with R2 [31]. Therefore, in this study, all excipients showed an increase in R7 of which not only Vaseline and foam increased it the most, but also increased R2 values. For this reason, these two excipients should be included in anti-aging products; however, the application of the excipient was performed once and during a short period of time, therefore, future experiments should be performed to verify this hypothesis. Nevertheless, considering the organoleptic properties of the excipient, the viscosity of Vaseline could be bothersome for consumers. In that way, the Beeler excipient may be a better option because it confers greater hydration and a higher increase in elasticity.

The data presented above should be considered by the pharmaceutical industry for their future topical drugs. For example, generic topical drugs must have bioequivalence with the reference listed drug (RLD). When a generic product does not have the same excipients as RLD, it must prove its qualitative and quantitative properties and the similarity in in vitro release tests (IVR). If they do not meet the qualitative and quantitative requirements, the excipients must be tested in IVR trials to demonstrate that the safety and efficacy of the product have not changed and the inert property of the excipient is conserved. In the failure to comply with IVR trials, the generic topical drug is not eligible for biowaiver and requires in vivo bioequivalence studies, increasing its final price for consumers [32]. For this reason, the choice of a proper excipient is of great importance, not only for its clinical implications, but its economic consequence for the pharmaceutical industry [33]. Further research is also needed to know if the effect is similar in females and males or in patients of different ages.

This study was subject to the following limitations: (1) The variability of skin homeostasis parameters is reliant on external conditions; however, to maximize outcome accuracy, all participants were measured in the same room, and the ambient conditions were also measured. (2) There are more excipients that could have been included in the study. For this reason, further studies should be conducted to show how excipients cross-interacted with skin homeostasis and active ingredients.

## 5. Conclusions

This study compares the impact of the following W/O, O/W emulsions: Beeler base, foam and Vaseline on skin barrier function. All excipients decreased erythema and increased elasticity. Vaseline decreased TEWL and increased elasticity. Foam also increased TEWL. Beeler showed the greatest hydration power.

Knowing the effect of each excipient on the skin could help to develop new topical treatments and creams that will release to improve absorption, hydration and dysbiosis with lower rates of epidermal disruption. Besides, these data could be helpful to better choose the excipients according to the pathology.

## Figures and Tables

**Figure 1 ijerph-19-16678-f001:**
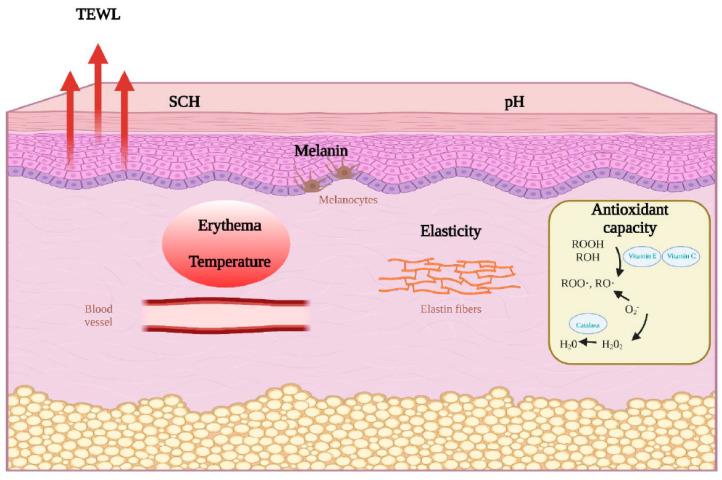
Skin barrier function parameters that can be evaluated non-invasively on the skin. SCH: stratum corneum hydration; TEWL: transepidermal water loss. This figure has been created using BioRender.com. It has been adapted from Trinidad Montero-Vilchez PhD work.

**Figure 2 ijerph-19-16678-f002:**
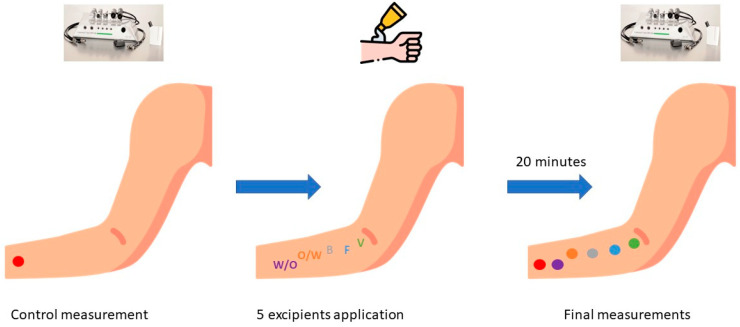
Graph summarizing the methods used to evaluate skin barrier function. B, beeler; F, foar; O/W, oil-in-water; V, Vaseline; W/O, water-in-oil. This figure was created using icons of flaticon.es.

**Figure 3 ijerph-19-16678-f003:**
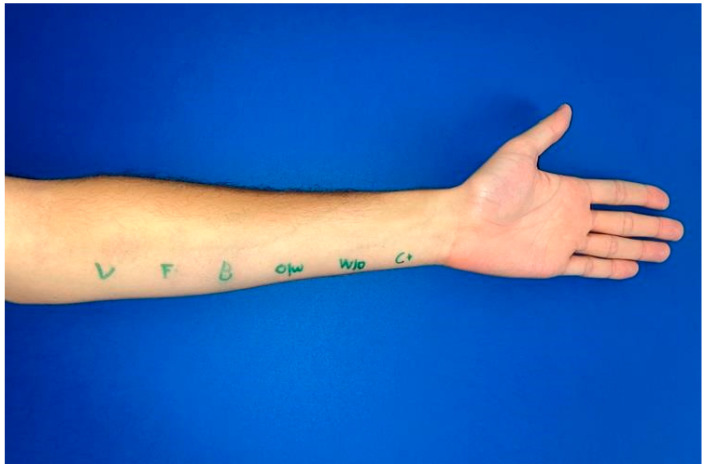
Labeled forearm where the creams were applied: Control, C+; water-in-oil, W/O; oil-in-water, O/W; Beeler, B; foam, F; Vaseline, V.

**Figure 4 ijerph-19-16678-f004:**
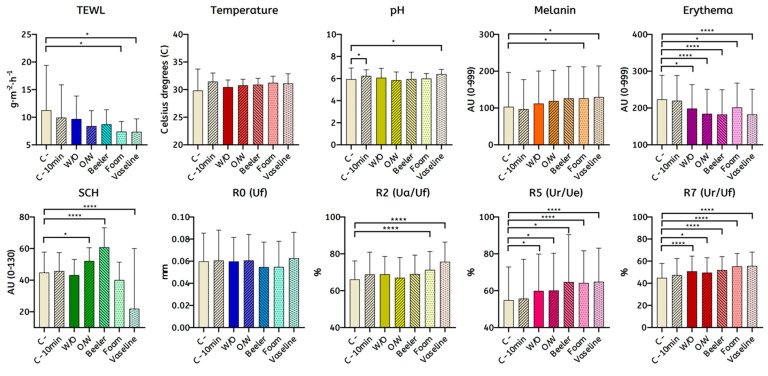
Changes in skin homeostasis testes in humans after 10 min of excipients treatment: the excipients W/O, O/W, Beeler, foam and Control were applied for a further 10 min before measuring TEWL, temperature, pH, melanin, erythema, SCH and elasticity values (R0, R2, R5, R7) with Tewameter^®^, Skin-Thermometer ST 500, Skin-pH-Meter PH 905, TM 300, Mexameter^®^ and Corneometer^®^ and Cutometer^®^ Dual MPA 580. Two-way ANOVA followed by an α-adjusted post hoc Bonferroni test were performed; * *p* < 0.05; **** *p* < 0.0001.

**Table 1 ijerph-19-16678-t001:** Sociodemographic characteristics at baseline.

Age (Years) (Mean (SD))		32 (±14.3)
Gender n (%)		
	Male	27 (45%)
	Female	33 (55%)
Phototype n (%)		
	II	15 (25.4%)
	III	35 (59.3%)
	IV	7 (11.9%)
	V	1 (1.7%)
	VII	1 (1.7%)
Smokers n (%)		10 (16.7%)
Drinkers n (%)		38 (63.3%)
Education n (%)		
	University graduate	55 (91.7%)
	Vocational training	4 (6.7%)
Domicile n (%)		
	Urban	51 (85%)
	Rural	8 (13.3%)
Ocupation n (%)		
	Student	30 (50.8%)
	Nurse	11 (18.6%)
	Doctor	10 (16.9%)
	Auxiliary nurse	4 (6.8%)
	Nursing residency	1 (1.7%)
	Orderly	1 (1.7%)
	Administrative staff	1 (1.7%)
	Unemployed	1 (1.7%)
Skin Hydration (yes) n (%)		33 (55%)
Photoprotector n (%)		
	Always	14 (23.3%)
	Sometimes	28 (46.7%)
	Never	17 (28.3%)

For quantitative measures, mean and SD were calculated whereas for quantitative measures on percentage and n are showed.

## Data Availability

The data presented in this study are available from the corresponding author on request.

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
