# Peer review of "The Assessment of Skin Homeostasis Changes after Using Different Types of Excipients in Healthy Individuals"

_ijerph, 2022, doi:10.3390/ijerph192416678_

Round 1
Reviewer 1 Report (New Reviewer)
1. The introduction provide sufficient background and include all relevant references. But they can add some diagrams about parts of skin. They explained natural moisturizing factor 47 (NMF), transepidermal 48 water loss (TEWL), SC hydration (SCH), elasticity, pH etc. Why do you use some histopathological examination?
2. Healthy individuals aged between 18 and 65 years old. What is your opinion about the features of young people and old people? Does it any differences or not?
3. I can not understand the relations of sociodemographic parameters and clinical characteristics of baseline. Writers should explain much details of correlations in the text.
Author Response
- The introduction provide sufficient background and include all relevant references. But they can add some diagrams about parts of skin. They explained natural moisturizing factor 47 (NMF), transepidermal 48 water loss (TEWL), SC hydration (SCH), elasticity, pH etc. Why do you use some histopathological examination?
Thank you for the comments. We have included a picture following your recommendation. We evaluate changes in skin barrier function parameters that can be measured non-invasively on the skin and do not perform histological examination of the skin to avoid take biopsy of healthy volunteers.
- Healthy individuals aged between 18 and 65 years old. What is your opinion about the features of young people and old people? Does it any differences or not?
It has been described that skin barrier function changes with age and external factors. Skin changes fundamentally in the extreme stages of life so we did not include these ages. Moreover, we are comparing the impact of several excipients in the same individual meaning that a participant is his own control. In that way we may not expect that age influence on our results.
- I can not understand the relations of sociodemographic parameters and clinical characteristics of baseline. Writers should explain much details of correlations in the text.
Despite each subject was his own control, we also evaluated if there could be differences between sociodemographic characteristics but we did not find any. Following your recommendations we have included this aspect in the text.
Reviewer 2 Report (New Reviewer)
The manuscript „THE ROLE OF EXCIPIENTS ON EPIDERMAL BARRIER FUNCTION“ presents potentially useful results for further work and practice in various fields and medical branches.
However, I suggest to change the title (to be more precise), for instance it could be “The assessment of skin homeostasis changes after using different types of excipients in healthy individuals“.
MATERIALS AND METHODS: Please correct the number and the text. Namely, in this text it is written:
2.2.1 Inclusion criteria
· Healthy individuals aged between 18 and 65 years old. 3.2.2 Exclusion criteria
· Patients having a diagnosticated skin diseases at the moment…
Also, the authors may present their methods in one figure to better insight into the methods for the reader. Also, it is needed to correct „vasleine“, „diagnosticated“ and check English in all manuscript.
RESULTS: The results are presented in one valuable table and one figure. It is difficult to follow the text because there are many parameters and I suggest adding a table with the presentation of crucial results by each study parameter.
In DISCUSSION, it could be useful to mention some more experience obtained by some additional studies/authors, if possible.
REFERENCES: It is needed to add some years for references which were missed, because they are not presented for some references. Also, if possible, the authors may add some recent references.
However, since I am a dermatologist, I can comment on the text as a clinician and I think that it would be useful to check the obtained data by a pharmacist.
Author Response
The manuscript „THE ROLE OF EXCIPIENTS ON EPIDERMAL BARRIER FUNCTION“ presents potentially useful results for further work and practice in various fields and medical branches.
Thank you for the comments
However, I suggest to change the title (to be more precise), for instance it could be “The assessment of skin homeostasis changes after using different types of excipients in healthy individuals“.
We have changed the title following your recommendations.
MATERIALS AND METHODS: Please correct the number and the text. Namely, in this text it is written:
2.2.1 Inclusion criteria
- Healthy individuals aged between 18 and 65 years old. 3.2.2 Exclusion criteria
- Patients having a diagnosticated skin diseases at the moment…
We have checked the number and the text.
Also, the authors may present their methods in one figure to better insight into the methods for the reader.
Following your recommendation, we have included a figure that summarize the methods.
Also, it is needed to correct „vasleine“, „diagnosticated“ and check English in all manuscript
We have checked these words and reviewed the English.
RESULTS: The results are presented in one valuable table and one figure. It is difficult to follow the text because there are many parameters and I suggest adding a table with the presentation of crucial results by each study parameter.
Thank you for the comment. We have included the table as supplementary material (table S2).
In DISCUSSION, it could be useful to mention some more experience obtained by some additional studies/authors, if possible.
We have included more experience obtained by other authors as recommended.
REFERENCES: It is needed to add some years for references which were missed, because they are not presented for some references. Also, if possible, the authors may add some recent references.
We have reviewed the references and have included recent references.
However, since I am a dermatologist, I can comment on the text as a clinician and I think that it would be useful to check the obtained data by a pharmacist.
We agree with your comment. Both dermatologist and pharmacist should be concern about the role of excipients in skin barrier function. In that way, this research was conducted by dermatologists and pharmacists.
Round 2
Reviewer 1 Report (New Reviewer)
Thank you very much about reorganisation.
Reviewer 2 Report (New Reviewer)
The authors accepted the recommendations and corrected the text.
This manuscript is a resubmission of an earlier submission. The following is a list of the peer review reports and author responses from that submission.
Round 1
Reviewer 1 Report
This study is an intervention study and must be reported accordingly. If you accept, that the foam is a product format, but an emulsion in the end on the skin surface, the entire results and discussion sections must be changed. It is more than unlikely, that effects after 10 minutes (the water phase is still evaporating), are similar to long-term effects. The colour readings make no sense at all.
Reviewer 2 Report
Thank you for the revisions already done, however, the English language still needs substantial revision which would allow the reader to understand the study objectives, methods, results, and conclusions.
The introduction can be improved by correctly presenting the skin layers starting from the most superficial to those below.
Reviewer 3 Report
The current study investigated the effects of W/O, O/W emulsions; Beeler base, Foam, and Vaseline on the skin barrier function. This is interesting, as it could be valuable in developing or improving topical drug efficacies for skin diseases. However, the authors are expected to address some of the issues/questions.
1) The measurements were performed 10 minutes after the five bases, which should be considered as a short-term effect. However, the long-term effects observed in clinical practices or daily life use should be discussed.
2) Interestingly, there is a significant difference in erythema levels after the bases for 10 minutes. I am wondering if there is a difference between males and females. This could be discussed in the discussion section.
3) Figure 2 shows a significant difference in pH between the C group and C-10min groups. A better explanation is necessary regarding why this happens.